# Agent S2: A Compositional Generalist-Specialist Framework for Computer Use Agents

**Saaket Agashe,**\* **Kyle Wong,**\* **Vincent Tu,**\* **Jiachen Yang, Ang Li, Xin Eric Wang**
Simular Research

## Abstract

Computer use agents automate digital tasks by directly interacting with graphical user interfaces (GUIs) on computers and mobile devices, offering significant potential to enhance human productivity by completing an open-ended space of user queries. However, current agents face significant challenges: imprecise grounding of GUI elements, difficulties with long-horizon task planning, and performance bottlenecks from relying on single generalist models for diverse cognitive tasks. To this end, we introduce Agent S2, a novel compositional framework that delegates cognitive responsibilities across various generalist and specialist models. We propose a novel Mixture-of-Grounding technique to achieve precise GUI localization and introduce Proactive Hierarchical Planning, dynamically refining action plans at multiple temporal scales in response to evolving observations. Evaluations demonstrate that Agent S2 establishes new state-of-the-art (SOTA) performance on three prominent computer use benchmarks. Specifically, Agent S2 achieves 18.9% and 32.7% relative improvements over leading baseline agents such as Claude Computer Use and UI-TARS on the OSWorld 15-step and 50-step evaluation. Moreover, Agent S2 generalizes effectively to other operating systems and applications, surpassing previous best methods by 52.8% on WindowsAgentArena and by 16.52% on AndroidWorld relatively. Code available at https://github.com/simular-ai/Agent-S.

## 1 Introduction

Computer-use agents are autonomous AI agents that can directly interact with graphical user interfaces to complete user requests. They function as intelligent intermediaries between humans and their digital tools in the most intuitive way: direct keyboard and mouse control. They can be generally used across a host of applications and websites that humans utilize without the need for specific APIs and protocols. While there has been substantial recent progress in developing computer use agents (Agashe et al., 2024; Qin et al., 2025; Anthropic, 2024; OpenAI, 2025; Bonatti et al., 2024), it remains a largely unsolved problem, with the best-performing agents lagging significantly behind humans (e.g., around 40% performance gap on OSWorld (Xie et al., 2024)).

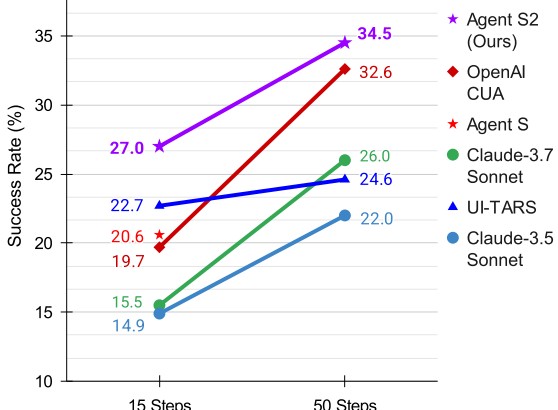

Figure 1: Agent S2 achieves new SOTA results (Success Rate) on computer use tasks on both 15-step and 50-step evaluation in OSWorld.

---

\*Equal contributions.

Current computer-use agents face three core limitations. First, they struggle to accurately ground textual descriptions of GUI elements to precise pixel-level coordinates. Second, they have difficulty handling long-horizon tasks, especially in the presence of background distractions, interruptions, and evolving user contexts and observations. Third, most methods rely solely on generalist models to perform diverse tasks such as planning, action generation, and grounding, leading to performance bottlenecks. While generalist models offer broad capabilities, they often underperform compared to specialist models in domain-specific subtasks, ultimately constraining the overall performance of computer-use agents.

To address these challenges, we introduce **Agent S2**, a compositional framework designed to delegate cognitive tasks across different generalist and specialist modules. First, Agent S2 tackles grounding bottlenecks via a novel Mixture of Grounding mechanism, where the agent reasons about subgoals and routes actions to specialized grounding experts for precise GUI localization across diverse applications. In addition, we propose a Proactive Hierarchical Planning method that dynamically adjusts and refines action plans at multiple temporal scales based on new observations. This enhances adaptability compared to passive or reactive planning methods, which either rigidly adhere to a predetermined script or only adjust after encountering execution failures. Overall, Agent S2 functions as a compositional and hierarchical system, distributing responsibilities across modules specialized in high-level reasoning, low-level execution, and fine-grained grounding, avoiding the limitations of monolithic approaches that rely solely on training or fine-tuning a single generalist model.

Our framework, Agent S2, achieves new state-of-the-art (SOTA) performance across multiple computer use benchmarks. Specifically, Agent S2 achieves 27.0% (↑ 18.9%)[1] and 34.5% (↑ 32.7%) on the OSWorld benchmark's (Xie et al., 2024) 15-step and 50-step evaluations, respectively, highlighting its effectiveness and scalability. Moreover, Agent S2 generalizes effectively to other benchmarks, achieving new SOTA results with 29.8% (↑ 52.8%) accuracy on WindowsAgentArena (Bonatti et al., 2024) and 54.3% (↑ 16.5%) accuracy on Android-World (Rawles et al., 2024b). Through comprehensive ablation studies, we highlight the improvements from our Mixture of Grounding strategy and the benefits of Proactive Planning over conventional reactive planning methods. We also analyze how scaling compute and time steps enhances performance and provide an extensive error analysis identifying current limitations and potential strategies for future improvements. Furthermore, our experiments validate that strategically composing generalist and specialist models, even when each is slightly suboptimal on its own, can outperform the best monolithic models.

We summarize our contributions as follows:

1. We introduce Agent S2: a new compositional, hierarchical framework for computer use agents that effectively delegates reasoning, execution, and grounding responsibilities across various generalist and specialist modules.
2. To address key limitations in existing computer use agents, we introduce Mixture of Grounding for resolving the grounding bottleneck and Proactive Hierarchical Planning for dynamic replanning in response to evolving observations and state changes.
3. We demonstrate that Agent S2 achieves state-of-the-art performance across multiple operating system benchmarks for Computer use and smartphone use tasks: OSWorld, WindowsAgentArena, and AndroidWorld.
4. Extensive ablation studies further demonstrate the effectiveness of the Mixture of Grounding and Proactive Hierarchical Reasoning for compositional frameworks. We also show a detailed thematic analysis of the emergent behaviors demonstrated by our agent with increased compute and time.

## 2 Background

**Computer Use Tasks and Benchmarks.** In computer use, agents interact with digital environments by executing desktop actions to fulfill user instructions. These tasks are inherently multimodal and can be formally described as a **Partially Observable Markov Decision**

---

[1]↑ represents a relative increase with respect to leading baselines UI-TARS and Claude Computer Use for OSWorld, UI-TARS for AndroidWorld, and NAVI Agent for WindowsAgentArena.

**Process (POMDP)**, defined as $\mathcal{M} = (\mathcal{S}, \mathcal{O}, \mathcal{A}, \mathcal{T}, \mathcal{R})$, where $\mathcal{S}$ is the state space (current state of the desktop), $\mathcal{O}$ is the observation space (instructions, screenshots, accessibility trees, etc.), $\mathcal{A}$ is the action space (e.g., click, type, etc.), $\mathcal{T} : \mathcal{S} \times \mathcal{A} \rightarrow \mathcal{S}$ is the state transition function, and $\mathcal{R} : \mathcal{S} \times \mathcal{A} \rightarrow [0, 1]$ is the reward function. For example, a user might request the agent to "change their default search engine". The agent must perceive the current screen – either through an image or an accessibility tree – and execute a sequence of actions to complete the given task.

Benchmarking multimodal agents for computer use has become a growing area of research. Early benchmarks (Deng et al., 2023; Rawles et al., 2023) utilized offline evaluations requiring agents to follow a specific action sequence for success, while recent online benchmarks use functional evaluation scripts. OSWorld (Xie et al., 2024) provides a structured environment for desktop control in Ubuntu, encompassing tasks like file management and document/image editing. WindowsAgentArena (Bonatti et al., 2024) extends these tasks to a parallelizable Windows environment, and AndroidWorld (Rawles et al., 2024a) includes mobile UI navigation tasks. Related benchmarks like Spider2-V Cao et al. (2024) also use functional evaluation, focusing on automating data science and engineering workflows. Other benchmarks like ScreenSpot (ByteDance, 2024) and VisualWebBench (Liu et al., 2024) focus explicitly on Visual Grounding in UI environments.

Recent works have approached these computer use tasks along two primary directions: (1) Training monolithic, generalist models on UI interactions and grounding data, and (2) Composing multiple models into hierarchical frameworks and creating a pipeline of planning, execution, and grounding.

**Monolithic Methods for Computer Use Agents.** Monolithic methods employ a single model to handle all aspects of computer use: planning, execution, and grounding. These systems require models to exhibit two critical abilities: (1) System-2 reasoning, which allows the generation of long-term plans and short-term actions, and (2) UI Grounding, which requires locating interactable elements by picking a precise coordinate from screenshots or a specific element from accessibility trees.

Several monolithic models post-trained on GUI tasks have emerged as native agents (Xu et al., 2024; Qin et al., 2025; Anthropic, 2024; Hong et al., 2023; Sun et al., 2025). Recent works explore strategies like Monte Carlo Tree Search (Yu et al., 2024), and learning from interactions (Su et al., 2025). AgentStore Jia et al. (2024) uses a MetaAgent that preselects application-specific agents(s) to perform a full task/subtask based on special tokens. Generalist approaches like Anthropic (2025) have even achieved state-of-the-art results on computer use tasks. Yet, monolithic methods have inherent drawbacks, as fine-tuning for specialization often diminishes broader capabilities (Yosinski et al., 2014; Luo et al., 2023), and assembling diverse, large-scale datasets for reasoning and visual grounding is expensive and time-intensive. Furthermore, relying on a single model may not optimally address the distinct needs of planning, execution, and grounding since these tasks can benefit from different model strengths.

**Hierarchical Methods for Computer Use Agents.** Hierarchical methods have gained popularity as a way to overcome the limitations of monolithic approaches, by decoupling the cognitive processes of planning, execution, and grounding. In hierarchical methods, at the high-level (time step $T$), a Manager $M$ generates a plan for instruction $I$, breaking it into coherent subgoals: $I = g_0, g_1, \ldots, g_N$. Each subgoal specifies an intermediate objective of the task. At the low-level (time step $t$), a Worker $W$ sequentially performs atomic actions (click, type, etc.) to iteratively complete each subgoal $g_i$: $g_i \rightarrow a_0, a_1, \ldots, a_t$.

Recent studies demonstrate hierarchical planning's effectiveness when combined with knowledge augmentation and continual learning (Agashe et al., 2024). Wu et al. (2024a) developed executor skill libraries complementing hierarchical decomposition. They perform Reactive planning for correction, updating subgoals only if a failure condition is seen. Wang & Liu (2024) uses a verification with hierarchical planning. Tan et al. (2024) builds a Multi-agent framework for distributing responsibilities to multiple models. Other frameworks have explored a modular approach by explicitly separating planning from

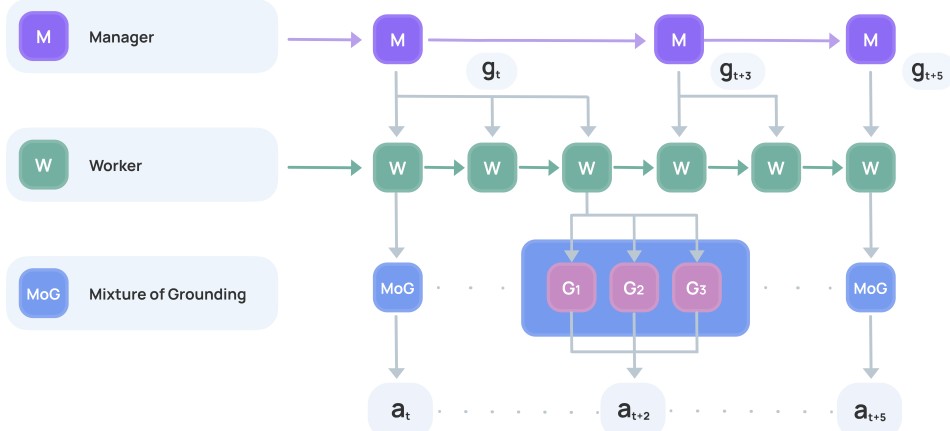

Figure 2: The Agent S2 framework. It composes *generalist* planning modules, Manager *M* and Worker *W*, with *specialist* grounding experts to complete complex, long-horizon computer use tasks. Please refer to Section 3 for a detailed explanation.

visual grounding. They train models (Wu et al., 2024b; Gou et al., 2024; Yang et al., 2024) for UI element grounding and then further pair them with general-purpose language models such as GPT-4o for planning. However, hierarchical systems also face key challenges. Workers managing both action generation and element grounding can become performance bottlenecks. Additionally, many planning methods are reactive and struggle to handle unexpected environmental changes, limiting overall robustness.

## 3 Agent S2: Compositional Grounding and Planning for Computer Use

Our *Agent S2* framework, as depicted in Figure 2, aims to address complex computer tasks by composing *generalist* hierarchical planning modules with *specialist* grounding modules. The key components of our framework are the Manager *M*, the Worker *W*, and the Mixture of Grounding Experts (MoG). The Manager *M* operates at a higher semantic space, decomposing a task into a list of high-level subgoals $g_t$. The worker *W* operates on a lower semantic level, generating natural language actions to complete the topmost subgoal. Agent S2 uses a Mixture of Grounding strategy, where the Worker *W* routes its actions to the correct specialist module among the grounding experts $\{G_i\}_1^N$, effectively addressing the grounding bottleneck. Additionally, Agent S2 utilizes a Proactive Hierarchical Planning strategy where the Manager *M* updates its list of remaining subgoals after the completion of each individual subgoal to adapt to newer observations, while the Worker *W* routes its action to a new expert after each action. Since computer use tasks require highly specialized domain knowledge about various applications and requests, Agent S2 also uses the knowledge base from Agent S (Agashe et al., 2024), featuring high-level task interaction experience, low-level subgoal interaction experience, and contextual web knowledge.

### 3.1 Mixture of Grounding

Operating user interfaces involves navigating a wide range of applications like menus, canvases, spreadsheets, etc. The inability to precisely and robustly locate various regions of interest on a screen forms a crucial bottleneck in current computer use agents. To efficiently handle precise UI element localization, Agent S2 introduces *Mixture of Grounding (MoG)*, which forms the Specialist part of our framework. Analogous to Mixture-of-Experts (Jacobs et al., 1991), the Worker *W* in our framework acts as a gating mechanism and routes each generated action to the correct grounding expert. The grounding expert then generates the pixel-level coordinates. This allows the worker to focus on reasoning while distributing the cognitive load of grounding to the appropriate expert.

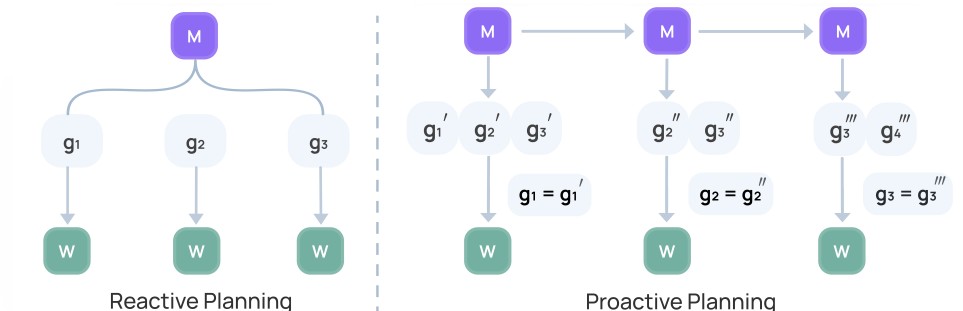

Figure 3: Comparison between Reactive and Proactive Planning. Proactive planning re-evaluates and updates the remainder of the plan after every subtask, while reactive planning adheres to a fixed plan and only revises it in response to subtask failures.

Formally, at each execution step, the Worker $W$ receives a subgoal $g_i$ alongside the latest environmental observation $o_t$. Policy $\pi_W$ generates atomic action $a_t$ necessary for executing $g_i$, each accompanied by a language descriptor specifying its target location. After deciding the atomic action $a_t$, the Worker delegates the grounding task to a corresponding expert among the following modules. The selection of the grounding expert is dynamically decided by the Worker module at each execution step.

**Visual Grounding Expert.** The visual grounding expert takes as input the current observation screenshot $o$ paired with a language description $d$ of a specific point in the image and generates the precise low-level coordinates $\langle x, y \rangle$ that represent $d$. This description-based visual grounding allows Agent S2 to rely solely on screenshots as input, eliminating the need for bulky accessibility trees and HTML. Most importantly, the visual grounding expert enables Agent S2 to act on any point on the screen rather than being restricted to selecting only high-level elements, greatly expanding the scope of possible interactions. Furthermore, the worker $W$ can progressively refine the descriptions it provides to the Visual Grounding expert to self-correct actions over time.

**Textual Grounding Expert.** While visual grounding models like UGround (Gou et al., 2024) and UI-TARS (Qin et al., 2025) have shown impressive precision, a class of problems that still poses a challenge is fine-grained text grounding, such as generating coordinates perfectly aligned with the edge of a word or sentence. To address this limitation, we use Optical Character Recognition (OCR), which is a conventional way of locating characters in textual documents and paragraphs. In addition to the current observation screenshot $o$, the Textual Grounding expert also takes two phrases $p_1$ and $p_2$ as input. $p_1$ and $p_2$ are the exact word sequences from the start and end of the span of interest. The Textual Grounding expert uses OCR to output the span coordinates $\langle x_{start}, y_{start} \rangle$, and $\langle x_{end}, y_{end} \rangle$.

**Structural Grounding Expert.** Another category of grounding bottleneck involves locating elements in spreadsheets and tabular content. Since spreadsheet cells can be stretched and squeezed to arbitrary sizes, and translating the table can change the starting position of the rows and columns, grounding on tabular data remains a significant challenge. To overcome this limitation and ensure precise grounding in tabular UI elements, the Structural Grounding expert takes a dictionary of $\langle$ "cell": "value" $\rangle$ mapping and programmatically updates the content of the corresponding cells. The structural grounding expert can take multiple cells, even entire rows, columns, or tables, as input and update them all at once, allowing both reliable and faster grounding for structured data.

## 3.2 Proactive Hierarchical Planning

Computer use tasks can often span long horizons involving multiple apps, screens, and a long series of observations. The initial state often contains partial information needed to address the user's query. Moreover, background apps and pop-ups introduce significant noise, and the susceptibility of Multimodal LLMs to UI noise (Ma et al., 2024) further complicates the problem. Therefore, Agent S2 incorporates Proactive Hierarchical Planning,

| Framework | Model | 15-step | 50-step |
|---|---|---|---|
| UI-TARS (Qin et al., 2025) | UI-TARS-72B-SFT | 18.7 | 18.8 |
| UI-TARS (Qin et al., 2025) | UI-TARS-72B-DPO | 22.7 | 24.6 |
| OpenAI CUA (OpenAI, 2025) | OpenAI Operator | 19.7 | 32.6 |
| Aria-UI (Yang et al., 2024) | GPT-4o | 15.2 | – |
| Aguvis-72B (Xu et al., 2024) | GPT-4o | 17.0 | – |
| Agent S (Agashe et al., 2024) | GPT-4o | 20.6 | – |
| Agent S2 **(ours)** | GPT-4o | **21.1** | **26.6** |
| Claude Computer Use (Anthropic, 2024) | Claude-3.5-Sonnet new | 14.9 | 22.0 |
| Agent S (Agashe et al., 2024) | Claude-3.5-Sonnet | 20.5 | – |
| Agent S2 **(ours)** | Claude-3.5-Sonnet new | **24.5** | **33.7** |
| Claude Computer Use (Anthropic, 2025) | Claude-3.7-Sonnet | 15.5 | 26.0 |
| Agent S2 **(ours)** | Claude-3.7-Sonnet | **27.0** | **34.5** |

Table 1: Success Rate (%) on OSWorld across different agent frameworks and foundation models. Agent S2 achieves new state-of-the-art results on OSWorld for both 15- and 50-step evaluations with multiple foundation models, such as Claude-3.5-Sonnet, Claude-3.7-Sonnet, and GPT-4o. All agents use only screenshots as input, except Agent S, which uses both the accessibility tree and screenshots.

which replans and reasons at both levels of hierarchy (Manager and Worker) over different temporal scales. Unlike reactive planning approaches, which only update their plans after failure (see Figure 3), proactive planning allows Agent S2 to update its plan after completing every subgoal, effectively adapting to evolving observations and recontextualizing the user query while maintaining context from previous subgoals to reduce susceptibility to noise.

At each high-level time step $T$, given a user instruction $I$ and the current observation $o_0$, the Manager $M$ generates a plan, which is a sequence of subgoals $\{g'_1, g'_2, g'_3, \ldots, g'_n\}$. The Worker $W$ then takes the first subgoal $g_1 = g'_1$ and begins executing it. To do this, at each low-level time step $t$, the Worker follows its policy $\pi_W$ to pick actions $a_t$ and routes the action to the appropriate grounding expert as explained in Section 3.1. After several low-level steps, the Worker concludes the subgoal $g'_t \to a_0, a_1, \ldots, a_t$ with either SUCCESS or FAILURE, returning control to the Manager. Then, the Manager takes the prior subgoals $\{g'_1, g'_2, g'_3, \ldots, g'_n\}$, the latest observation $o_t$, and the original instruction $I$ as input. The context from the previous subgoals allows the Manager to bootstrap and connect its thinking to the original task while allowing it to incorporate the new observations. Based on these prior subgoals and the latest observations, it generates a new set of subgoals $\{g''_2, g''_3, \ldots, g''_n\}$. The first subgoal from this updated list becomes the next Worker objective, $g_2 = g''_2$. This process continues as required, with Manager refining subgoals until instruction $I$ is resolved.

# 4 Experiments

## 4.1 Experimental Setup

**Benchmarks.** We run our main experiments on the OSWorld (Xie et al., 2024) benchmark consisting of 369 real-world computer use tasks across the following categories: OS, Office (LibreOffice Calc, Impress, Writer), Daily (Chrome, VLC Player, Thunderbird), Professional (VS Code and GIMP) and Workflow (tasks involving multiple apps). We further evaluate Agent S2 on WindowsAgentArena (Bonatti et al., 2024) with 154 tasks executed on the Windows operating system. To generalize beyond computer use, we test on the Android-World (Rawles et al., 2024b) benchmark with 116 Smartphone use tasks across 20 real-world Android applications. For ablation studies, we utilize a subset of OSWorld, consisting of 65 examples sampled from the OSWorld environment, stratified by categories.

**Baselines.** Across each benchmark, we mainly compare with screenshot-input baselines. For OSWorld, we compare our method with OpenAI CUA / Operator (OpenAI, 2025), Claude Computer Use (CCU) with 3.5-Sonnet and 3.7-Sonnet (Anthropic, 2024), and UI-

| Model | OS | Daily | Office | Professional | Workflow | Overall |
|---|---|---|---|---|---|---|
| GPT-4o | 50.00 | 30.70 | 18.97 | 51.02 | 14.93 | 26.62 |
| Claude-3.5-Sonnet (new) | **58.33** | 48.44 | **29.06** | 51.02 | 13.46 | 33.71 |
| Claude-3.7-Sonnet | 50.00 | **49.73** | 25.64 | **57.14** | **18.21** | **34.47** |

Table 2: Categorized Success Rate (%) of Agent S2 on the OSWorld 50-step evaluation. We report results with various MLLMs as Manager and Worker.

| Method | OS | Daily | Office | Professional | Workflow | Overall |
|---|---|---|---|---|---|---|
| Aria-UI (GPT-4o) | 25.00 | 25.71 | 9.58 | 20.41 | 8.55 | 15.15 |
| Agent S2 (Qwen2.5-VL-72B-Instruct) | 50.00 | 29.42 | 5.98 | 42.86 | 4.49 | 18.29 |
| Agent S (GPT-4o)* | **45.84** | 27.06 | 13.00 | 36.73 | 10.53 | 20.58 |
| Agent S2 (GPT-4o) | 37.50 | 25.57 | 11.96 | 48.98 | **10.89** | 21.12 |
| Agent S2 (Claude-3.5-Sonnet (new)) | **45.84** | **35.82** | 15.52 | 57.14 | 5.00 | 24.50 |
| Agent S2 (Claude-3.7-Sonnet) | 41.67 | 35.62 | **20.51** | **65.31** | 5.94 | **27.04** |

Table 3: Categorized Success Rate (%) on the OSWorld 15-step evaluation across various agent configurations.

*Uses Atree in addition to screenshot input.

TARS-72B-DPO (Qin et al., 2025). To standardize the evaluation and test for scalability, we show our results at both 15-step and 50-step evaluation. For WindowsAgentArena, we compare with the Navi Agent (Bonatti et al., 2024) + Omniparser (Lu et al., 2024). Notably, this result uses both the accessibility tree and screenshot as input, while we only require the screenshot. Lastly, for AndroidWorld, we compare with UI-TARS-72B-SFT (Qin et al., 2025) and GPT-4o + Aria-UI (Yang et al., 2024).

**Implementation Details.** For the Mixture of Grounding experts, Agent S2 uses UI-TARS-72B-DPO as the visual grounding expert, Tesseract OCR (OCR, 2025) as the textual grounding expert, and Universal Network Objects (UNO) (Unotools, 2025) interface as the structural grounding expert. The backbone models evaluated include Claude-3.7-Sonnet, Claude-3.5-Sonnet (new), and GPT-4o (Specifically, versions claude-3-7-sonnet-20250219, claude-3-5-sonnet-20241022, and gpt-4o-2024-08-06). The best results on Computer use tasks are from evaluations with Claude-3.7-Sonnet, with its thinking mode enabled to fully leverage the reasoning capabilities in long-horizon tasks. For evaluations on OSWorld and WindowsAgentArena, test tasks that reach the allocated step limit before the agent signals task completion are considered failures. For AndroidWorld, we use Agent S2 in a worker-only setting due to shorter horizon tasks. In all environments, we use screenshots and action history as input.

## 4.2 Main Results

**OSWorld.** Table 1 presents Agent S2's performance on the OSWorld Benchmark. Agent S2 with Claude-3.7-Sonnet or Claude-3.5-Sonnet (new) achieves new SOTA, outperforming all other results on both 15-step and 50-step evaluations. Notably, Agent S2 with Claude-3.5-Sonnet (new) relatively outperforms Claude Computer Use with Claude-3.7-Sonnet by 58.1% on 15-step and 29.6% on 50-step evaluations, illustrating the advantages of modular, hierarchical frameworks over monolithic generalist modules in long-horizon tasks like computer use. Table 2 provides a further detailed breakdown of performance across OSWorld's categories for 50-step evaluation, where Agent S2 demonstrates high effectiveness on OS, Daily, and Professional tasks across all backbone models. It also delivers competitive performance on Office tasks, a historically challenging category (Agashe et al., 2024; Su et al., 2025). Surprisingly, Agent S2 with Claude-3.5-Sonnet (new) surpasses the Claude-3.7-Sonnet variant in the Office category. Closer analysis indicates that it relies on Textual and Structural Grounding experts almost twice as often, highlighting the benefits of Mixture of Grounding. Figure 4 shows an example of Agent S2 in action, where it resorts to alternate grounding expert and then replans based on new observation. Table 3 compares the 15-step performance of Agent S with various underlying based models and against previous baselines demonstrating consistently better performance.

| Method | Model | Office | Web | Win. System | Coding | Media & Video | Win. Utils | Overall |
|--------|-------|--------|-----|-------------|--------|---------------|------------|---------|
| Agent S | GPT-4o | 0.0 | 13.3 | 45.8 | 29.2 | 19.1 | 22.2 | 18.2 |
| NAVI | GPT-4V-1106 | 0.0 | **27.3** | 33.3 | 27.3 | **30.3** | 8.3 | 19.5 |
| Agent S2 (Ours) | Claude-3.7-Sonnet | **7.0** | 16.4 | **54.2** | **62.5** | 28.6 | **33.3** | **29.8** |

Table 4: Success Rate (%) on the WindowsAgentArena test set (within 15 steps). Note that both Agent S and NAVI use screenshots and accessibility trees, while our agent only takes screenshots as the input. Agent S2 sets new SOTA on WindowsAgentArena.

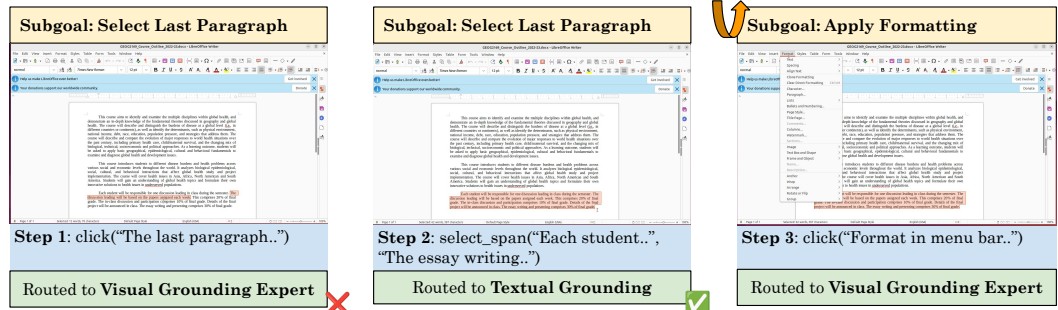

Figure 4: Agent S2 attempts to use Visual Grounding Expert to select a paragraph, then self-corrects and uses Textual Grounding Expert for span selection. After completing the subgoal, it replans from the new state and starts working on a new subgoal.

**WindowsAgentArena.** Agent S2's success also carries over to the WindowsAgentArena benchmark, which features Computer use tasks in the Windows Operating system. Table 4 shows that Agent S2 significantly outperforms the previous best agent NAVI (Bonatti et al., 2024) by 52.8%. The performance improvement is consistent across 4/6 categories. Notably, it is able to perform well on Windows-specific tasks, demonstrating generalization across operating systems.

### 4.3 Ablation Study

**Mixture of Grounding improves subtask completion rate and thus the overall success.** Figure 5 illustrates the performance improvement provided by the Mixture of Grounding strategy, especially when provided more steps. Specifically, MoG increases success rate from 27.69% to 30.77% at shorter horizons (15 steps) and, more prominently, from 33.85% to 38.46% at longer horizons (50 steps). To further study the benefit of individual experts, we extract a subset of OSWorld examples where the agent routes actions to the Textual or Structural Grounding expert. We then re-evaluate those examples without the corresponding expert to measure their impact on successful subtask completion. When removing the textual grounding expert, the subtask success rate drops from 70.6% to 65.2%, and when removing the structural grounding expert, the subtask success rate drops from 73.7% to 69.4%.

Visual Grounding forms the foundation of screenshot-only agents and is used in every single example task by our framework. We conduct a 15-step evaluation of the various grounding models as our Visual Grounding expert, as reported in Figure 6. In general, incorporating a specialist model that has better UI grounding capabilities into a broader framework yields better performance on computer use tasks. More importantly, we observe that smaller specialist models, such as UI-TARS-7B-DPO and UGround-V1-7B, can outperform large generalist models like Claude-3.7-Sonnet when employed within a modular framework that balances cognitive load effectively.

**Proactive planning enables self-correction and contextualization with new observations.** Our ablation study in Figure 5 also demonstrates the efficacy of proactive planning, revealing a performance improvement of +4.62% at 15 steps and +6.15% at 50 steps compared to reactive planning. This increase in success rate confirms that proactive hierarchical planning substantially enhances task completion by adapting to evolving observations.

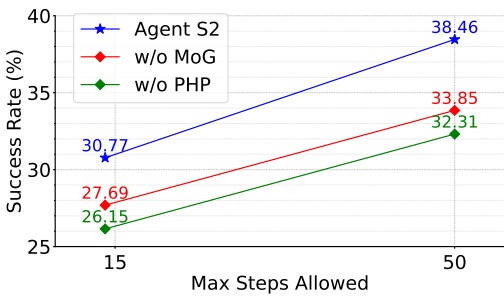

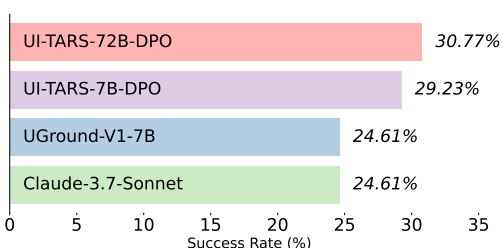

Figure 5: Ablation study on Mixture of Grounding (MoG) and Proactive Hierarchical Planning (PHP).

Figure 6: 15-step performance of Agent S2 with different visual grounding models. Small open-sourced models can outperform large close-sourced reasoning models.

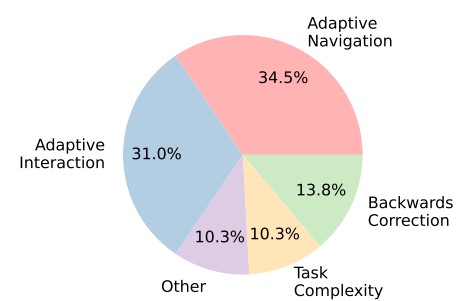

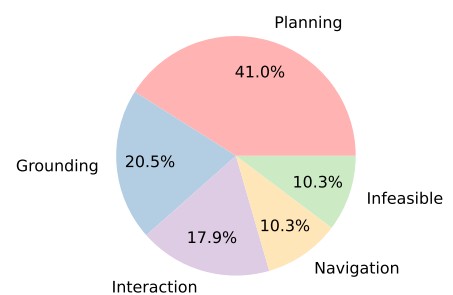

Figure 7: Contributing factors of 15-step to 50-step scaling success.

Figure 8: Failure categories on a subset of the OSWorld benchmark.

Similar to the improvement brought about by the Mixture of Grounding strategy, Proactive Hierarchical Planning is also more beneficial with more time steps.

**Agent S2 scales with more compute and steps.** To gain insight into the specific behaviors that enable Agent S2 to scale at test-time, we extract the subset of 29 OSWorld examples where Agent S2 fails on a 15-step evaluation but succeeds on a 50-step evaluation. As shown in Figure 7, our analysis reveals four primary behaviors that improve Agent S2's performance with additional steps. First, we observe that the most common behaviors are (1) *Adaptive Navigation*, where the agent explores multiple approaches to find a certain element or navigate to a certain page, and (2) *Adaptive Interaction*, where the agent interacts with the same element or page in different ways. These two behaviors show that the modular Agent S2 framework self-corrects and explores alternative approaches during inference, owing to its Proactive Hierarchical Planning abilities. Furthermore, it also intersperses various grounding experts to refine its interactions with the UI, as demonstrated in Figure 4. Other major contributing factors include: (3) *Backward Correction*, where the agent corrects small errors or missing interactions from previously completed subgoals while solving the current subgoal, which is promoted by the contextualization of user queries with new observations during Proactive Planning; and (4) *Task Complexity*, which includes tasks where a perfectly optimal agent or human would require more than 15 steps to complete. This analysis further solidifies the role of Proactive Hierarchical Planning and Mixture of Grounding in enabling our agent to improve with more time steps.

## 4.4 Error Analysis

Figure 8 displays the frequency of each failure type and offers insight into the current bottlenecks of Agent S2. We conduct error analysis on a subset of 65 samples from OSWorld through qualitative evaluation. For each example, we report the theme of the first major cause of failure. We observe the following failure modes: (1) *Planning failures*, where the manager formulates an inadequate plan, typically due to inaccurate/noisy subtask

information or misalignment with the task requirements. (2) *Grounding failures*, where the grounding expert produces inaccurate coordinates for the provided language description. (3) *Interaction failures*, where the worker is unable to successfully manipulate an element, reflecting a lack of domain knowledge on GUI interactions. (4) *Navigation failures*, where the worker struggles to find a certain element, suggesting deficiencies in layout understanding and navigation. (5) *Infeasible tasks*, for which the agent is unable to predict this infeasibility. Although previous works (Agashe et al., 2024; Xie et al., 2024) report grounding as a main cause of failure, we observe that Agent S2 maintains a relatively lower rate of grounding errors, while planning failures are now the most frequent. Furthermore, interaction and navigation failures are less common, which strengthens the findings from Figure 7 that Agent S2 adapts over longer horizons through test-time exploration.

## 4.5 Generalization to Smartphone Use

| Method | Base Model | SR (%) |
|---|---|---|
| Qwen2–VL–2B + InfiGUIAgent (Liu et al., 2025) | Qwen2–VL–2B | 9.0 |
| GPT–4 Turbo + AndroidWorld (Rawles et al., 2024b) | GPT–4 Turbo | 30.6 |
| GPT–4o + Ponder&Press (Wang et al., 2025) | GPT–4o | 34.5 |
| GPT–4o + UGround (Gou et al., 2024) | GPT–4o | 44.0 |
| GPT–4o + Aria–UI (Yang et al., 2024) | GPT–4o | 44.8 |
| UI–TARS–72B–SFT (Qin et al., 2025) | UI–TARS–72B | 46.6 |
| Agent S2 (Ours) | GPT–4o | **54.3** |

Table 5: Success rate (SR) comparison on the AndroidWorld benchmark for smartphone use.

We also perform a generalization study of Agent S2 on AndroidWorld (Rawles et al., 2024a) for smartphone use. Table 5 shows that Agent S2 outperforms the previous state-of-the-art method by a large margin of 16.5% relatively, which validates the strong generalizability and modularity of Agent S2.

## 5 Conclusion

We introduce Agent S2, a compositional framework integrating generalist and specialist models for high-level reasoning, low-level execution, and detailed grounding. Our Mixture of Grounding approach enlists a team of experts for precise grounding across diverse applications, while Proactive Hierarchical Planning refines plans and contextualizes observations based on user instructions. We show that Agent S2 achieves state-of-the-art performance on two computer use benchmarks and one smartphone use benchmark. We also perform ablation studies and error analysis to highlight the roles of each component in our system.

## Acknowledgments

We extend our sincere thanks to Tianbao Xie, Yujia Qin, Shihao Liang, and Zhiyong Wu for their engaging discussions on computer use agents.

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

## A    Domain Specific Results on OSWorld

| Model | OS | Gimp | Code | TB | Writer | Calc | Impress | Chrome | VLC | Multiapps | Overall |
|---|---|---|---|---|---|---|---|---|---|---|---|
| Claude-3.5-Sonnet | 58.33 | 38.46 | 65.22 | 60.00 | 43.48 | 25.53 | 25.53 | 41.19 | 57.87 | 13.46 | 33.71 |
| Claude-3.7-Sonnet | 50.00 | 50.00 | 65.22 | 73.33 | 34.77 | 25.53 | 21.28 | 41.19 | 51.99 | 18.21 | 34.47 |
| GPT-4o | 50.00 | 42.31 | 60.87 | 33.33 | 34.77 | 10.64 | 19.57 | 28.15 | 35.29 | 14.93 | 26.62 |

Table 6: Success rate (%) of Agent S2 on 50 step evaluation in OSWorld, divided by domains: OS, GIMP, VS Code, Thunderbird, LibreOffice Writer, LibreOffice Calc, LibreOffice Impress, Chrome, VLC, and Multiapps. We also report performance using different models as the Manager and Worker backbones.

| Method | OS | Gimp | Code | TB | Writer | Calc | Impress | Chrome | VLC | Multiapps | Overall |
|---|---|---|---|---|---|---|---|---|---|---|---|
| Aria-UI (GPT-4o) | 25.00 | 19.23 | 21.74 | 26.67 | 8.70 | 4.26 | 15.32 | 23.80 | 30.06 | 8.55 | 15.15 |
| Agent S (GPT-4o) | 45.84 | 23.08 | 52.17 | 40.00 | 30.42 | 2.13 | 15.34 | 21.74 | 30.06 | 10.53 | 20.58 |
| Agent S2 (GPT-4o) | 37.50 | 50.00 | 47.83 | 26.67 | 21.73 | 8.51 | 10.64 | 23.80 | 29.41 | 10.89 | 21.12 |
| Agent S2 (Claude-3.5-Sonnet) | 45.84 | 50.00 | 65.22 | 40.00 | 30.42 | 17.39 | 6.38 | 36.84 | 29.41 | 5.00 | 24.50 |
| Agent S2 (Claude-3.7-Sonnet) | 41.67 | 61.54 | 69.57 | 33.33 | 34.77 | 19.15 | 14.89 | 34.67 | 40.22 | 5.94 | 27.04 |

Table 7: Success rate (%) of Agent S2 and other baselines on 15 step evaluation in OSWorld, divided by domains. We also report performance using different models as the Manager and Worker backbones.

## B    App Specific Results on WindowsAgentArena

| Chrome | Edge | Code | Notepad | Lib_Calc | Settings | Win_Calc | Clock | Paint | File | Writer | VLC | Overall |
|---|---|---|---|---|---|---|---|---|---|---|---|---|
| 17.08 | 15.38 | 62.50 | 50.00 | 4.17 | 100.00 | 0.00 | 50.00 | 33.33 | 42.11 | 10.53 | 28.57 | 29.81 |

Table 8: Success rate (%) of Agent S2 with Claude-3.7-Sonnet on WindowsAgentArena, divided by apps: Chrome, Microsoft Edge, VS Code, Notepad, LibreOffice Calc, Settings, Windows Calc, Clock, Microsoft Paint, File Explorer, LibreOffice Writer, VLC Player.

## C    Agent S2 Action Space

To make it easier for the agent to interact with the environment, we create a structured action space using a function-calling inspired interface that streamlines the action selection and parameter specification process. Table 9 summarizes each action type, along with their specific parameters and a brief description of their usage.

| Agent Action | Action Details | |
|---|---|---|
| | Description | Arguments |
| click | Click on an element. | *element_description, num_clicks, button_type, hold_keys* |
| type | Type text into an element. | *element_description, text, overwrite, enter* |
| scroll | Scroll within an element. | *element_description, clicks, shift* |
| hotkey | Press a hotkey combo. | *keys* |
| hold_and_press | Hold keys and press others. | *hold_keys, press_keys* |
| drag_and_drop | Drag and drop between elements. | *element_description_1, element_description_2, hold_keys* |
| save_to_knowledge | Saves data to a per-task memory. | *text* |
| switch_applications | Switch to another app. | *app_name* |
| highlight_text_span | Highlights a text span. | *starting_phrase, ending_phrase* |
| set_cell_values | Sets tabular cell values. | *cell_values, app_name, sheet_name* |
| wait | Wait for some time. | *time* |
| done | Mark subtask as success. | None |
| fail | Mark subtask as failure. | None |

Table 9: Agent S2 action space, descriptions, and arguments.

# D    Case Studies on OSWorld

We present supplementary examples on OSWorld to demonstrate the efficacy of Proactive Planning and Mixture of Grounding. Some examples have less focal portions of the trajectory omitted in order to better highlight key behaviors of Agent S2. Figure 9 and Figure 10 showcase grounding experts in action, while Figure 11 displays Proactive Planning.

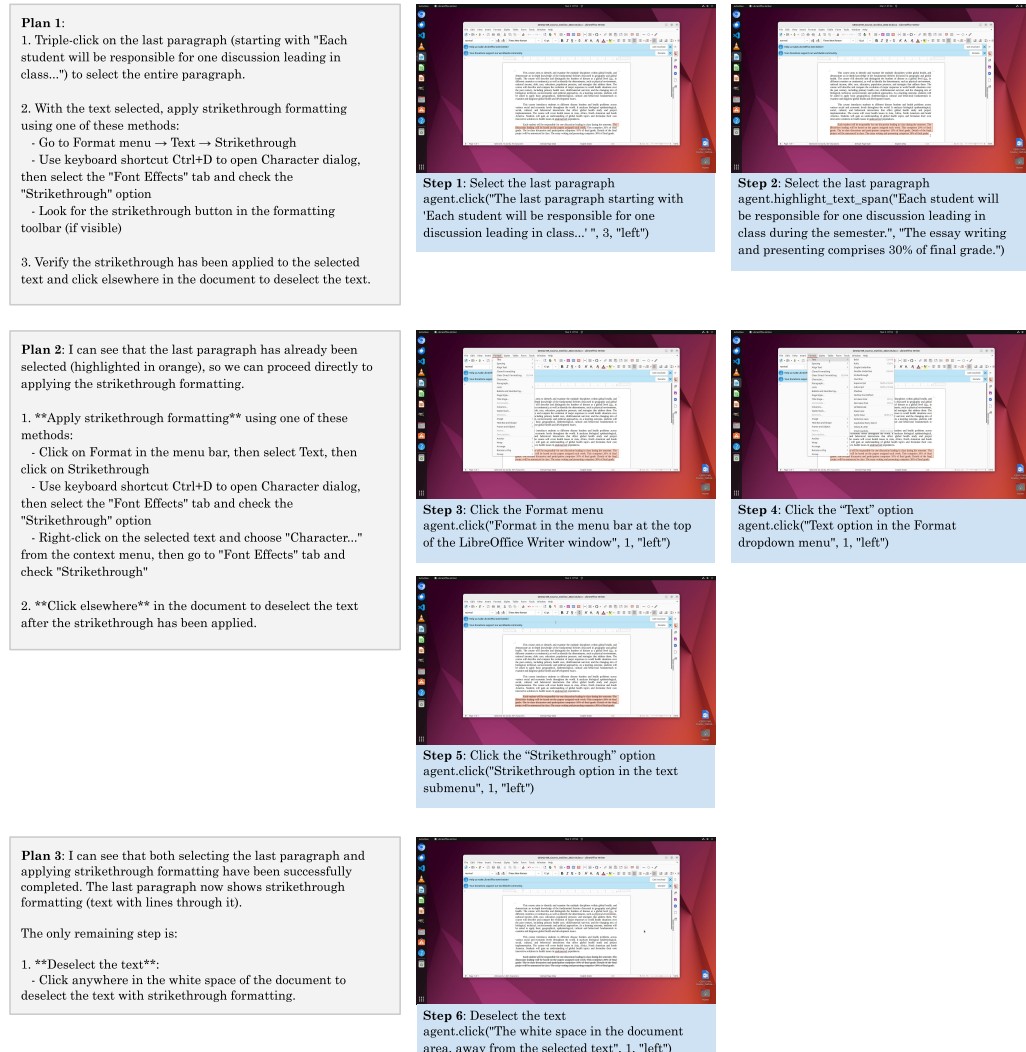

Figure 9: Agent S2 demonstrates textual grounding. The task instruction is: *I think the last paragraph is redundant so I want to add strike-through on words in the last paragraph. Can you do this for me?* Agent S2 intelligently utilizes the textual grounding expert to perform a precise word-level selection of text.

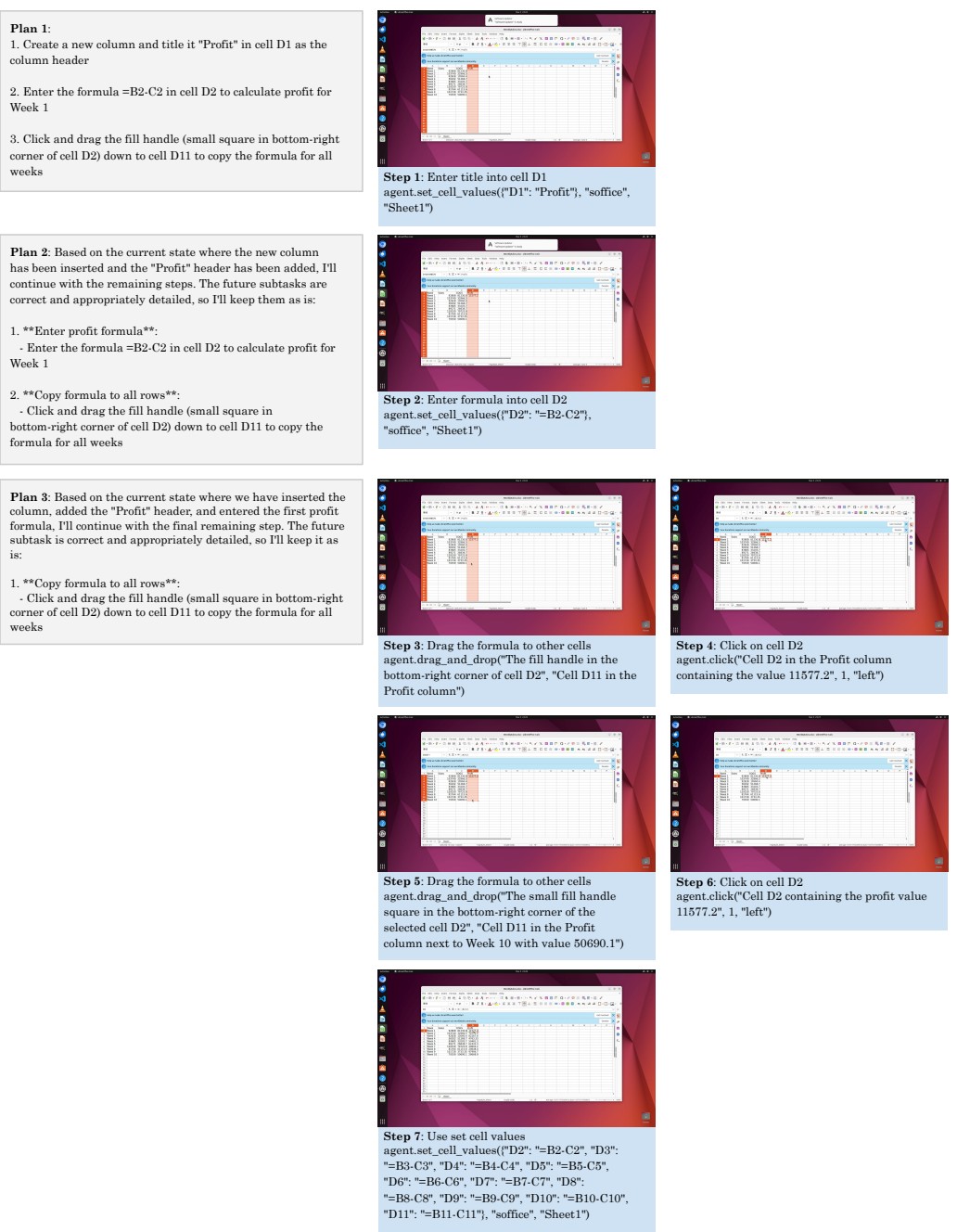

Figure 10: Agent S2 demonstrates structural grounding. The task instruction is: *Add a new column named "Profit" and calculate the profit for each week by subtracting "COGS" from "Sales" in that column.* Agent S2 adaptively interacts with the "Profit" column and chooses to use the structural grounding expert to perform cell-level operations.

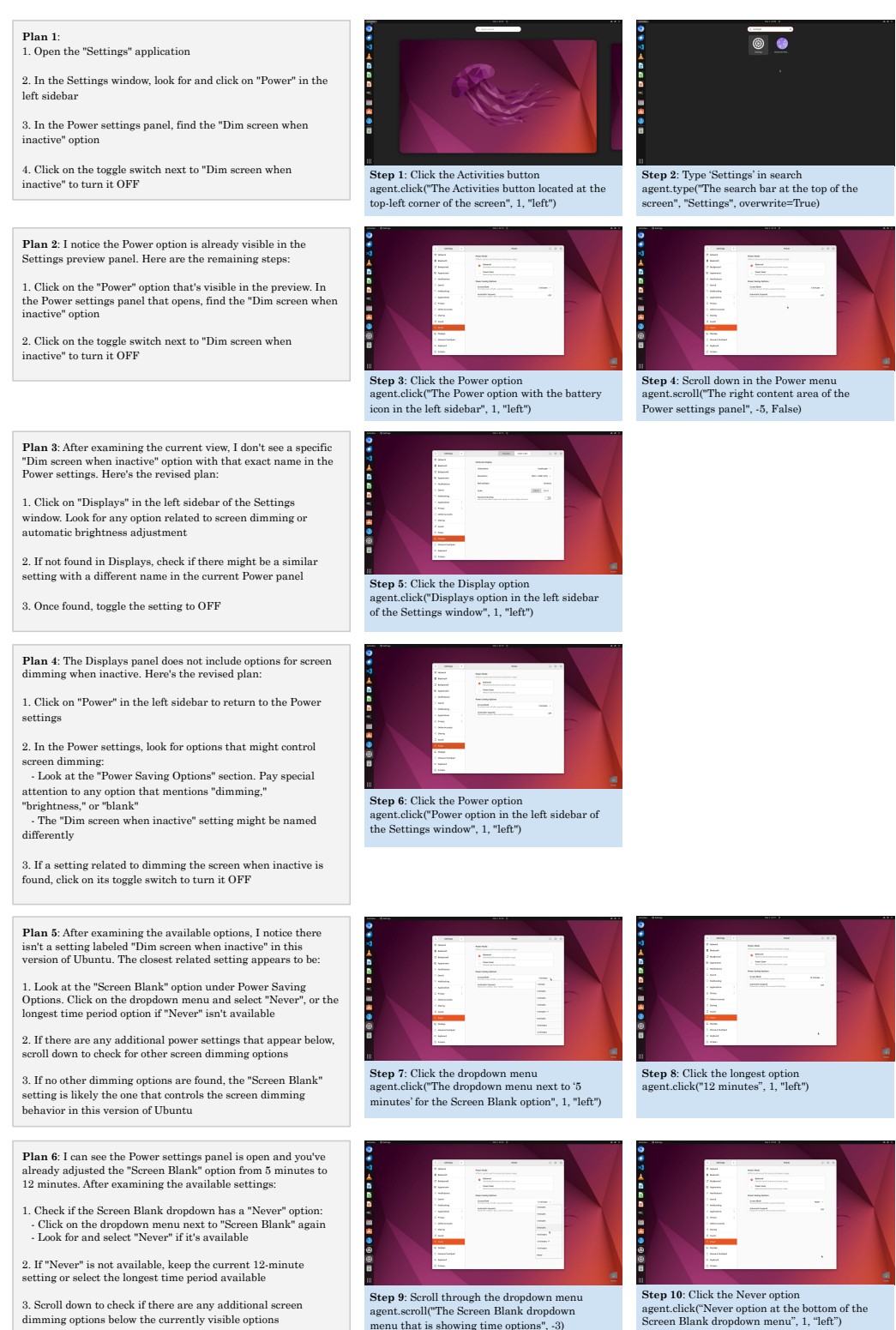

Figure 11: Agent S2 demonstrates replanning. The task instruction is: *Could you set the 'Dim screen when inactive' to off in settings?* Even though the 'Dim screen when inactive' option is not found in verbatim, Agent S2 proactively replans and reasons to find the correct setting.

# E    Case Studies on WindowsAgentArena

We present an example on WindowsAgentArena for a qualitative analysis in Figure 12.

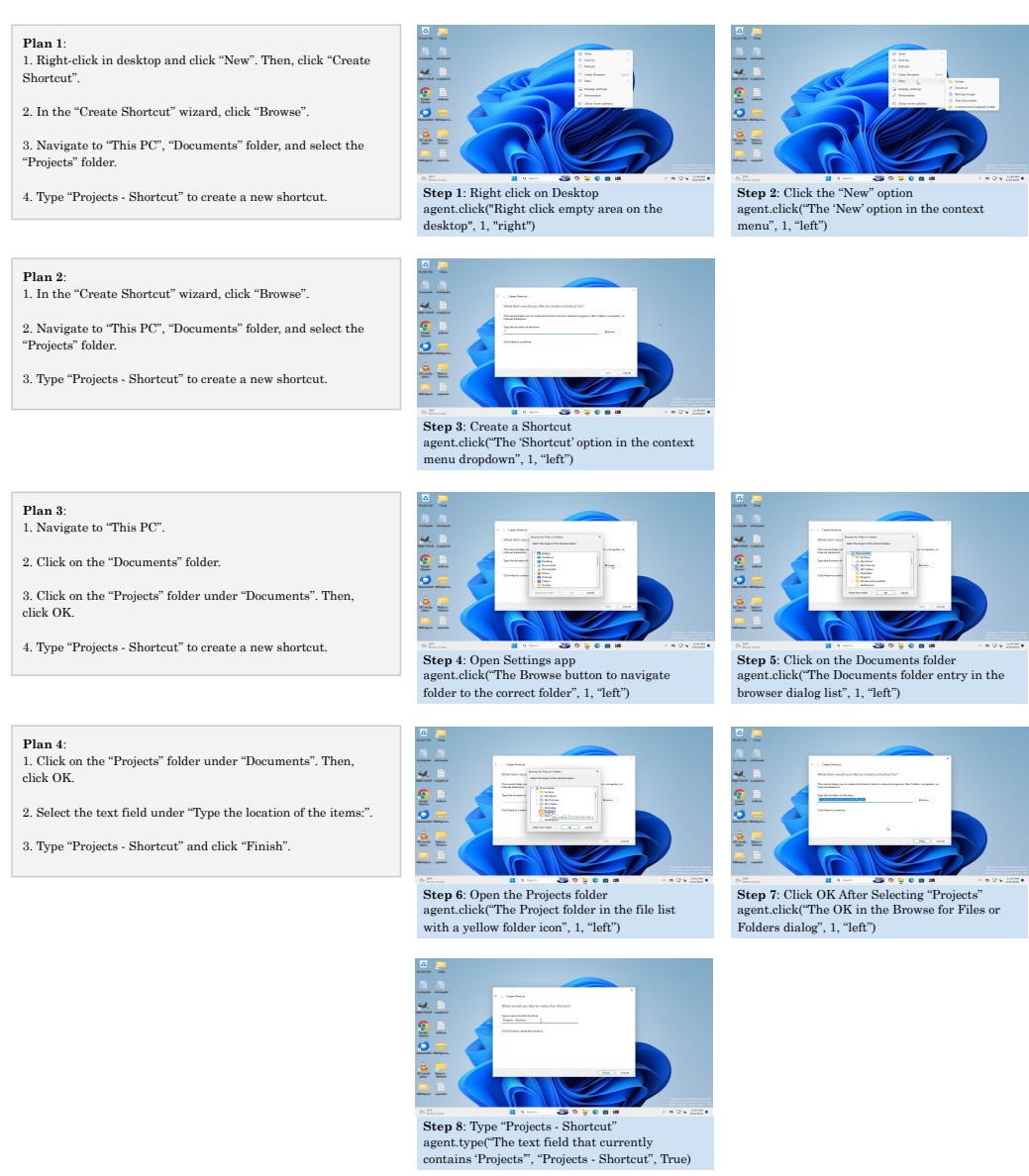

Figure 12: Agent S2 on the WindowsAgentArena environment. The task instruction is: *Create a shortcut on the Desktop for the folder named "Projects" that is located in the Documents folder. Name the shortcut "Projects - Shortcut"*. Through consistent replanning, Agent S2 is able to dynamically update its plan and revise its current subtask in more detail.

# F   Case Studies on AndroidWorld

We present an example on AndroidWorld for qualitative analysis in Figure 13.

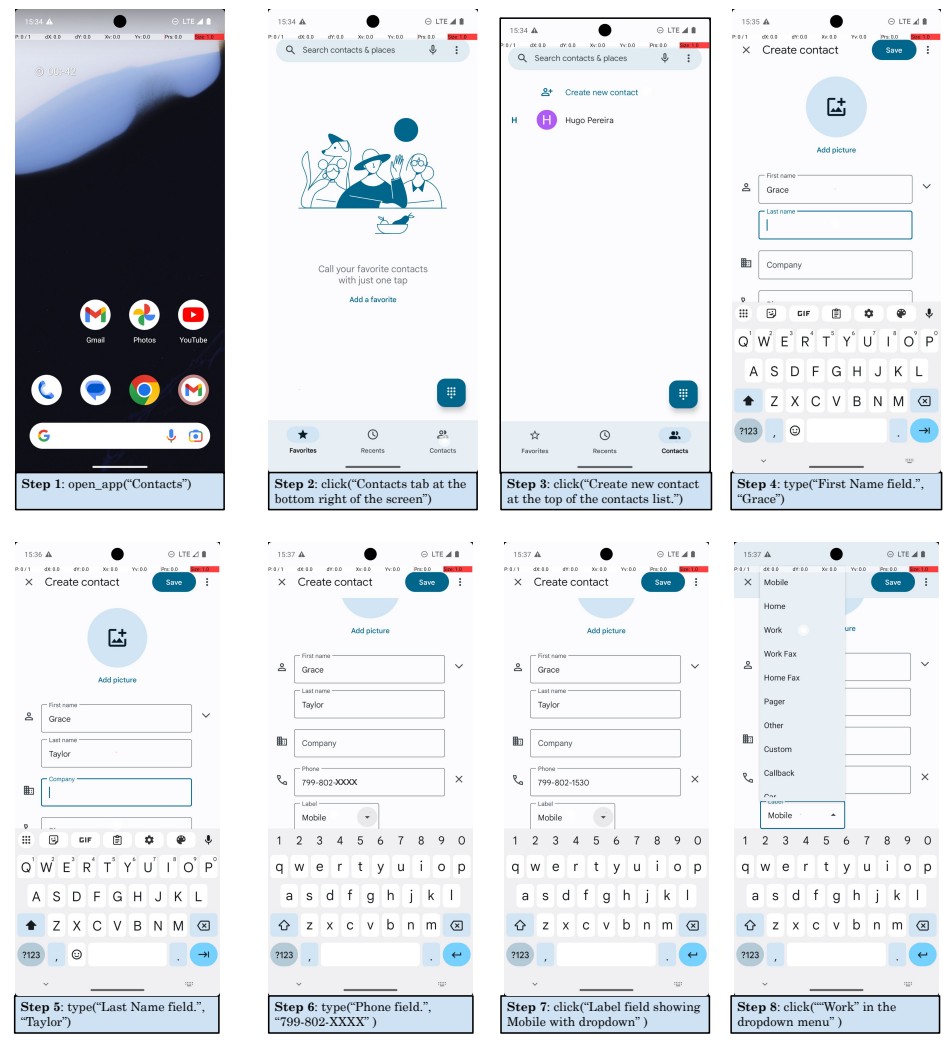

Figure 13: Agent S2 on the AndroidWorld mobile environment. It utilizes open, touch, and type interactions to complete the instruction *"Go to the new contact screen and enter the following details: First Name: Grace, Last Name: Taylor, Phone: 799-802-XXXX, Phone Label: Work"*.

