# OpenReview forum: "Agent S2: A Compositional Generalist-Specialist Framework for Computer Use Agents"
_colmweb.org/COLM/2025/Conference — COLM 2025_

### Official Review · Reviewer_Ygqb · 2025-05-11

**Rating:** 7
**Confidence:** 5
**Ethics Flag:** 1

**Summary:**

This paper proposes Agent S2, a compositional framework for computer use agents for GUI applications on desktops and mobile. Unlike single agents that rely on a single generalist model, Agent S2 decomposes cognitive responsibilities across generalist planning modules and specialist grounding experts. It introduces two key techniques: Mixture of Grounding, which enables precise GUI element localization by routing actions to the appropriate grounding expert, and Proactive Hierarchical Planning, which continuously updates task plans based on evolving observations. Agent S2 achieves good results on three dynamic benchmarks. It also shows strong some generalization and improved capability on long-horizon tasks.

**Questions To Authors:**

1. The routing strategy of Mixture-of-Grounding module lacks clarity, whether expert selection is fixed or dynamically switched (as in AgentStore), and whether this is based on hand-crafted heuristics?

I will discuss with the authors further.

**Reasons To Accept:**

1. The paper introduces a compositional generalist-specialist architecture that addresses some limitations of existing GUI agents. It propose Mixture of Grounding and Proactive Hierarchical Planning, which are practically useful for computer use scenarios.
2. Agent S2 achieves new SOTA on highly challenging OSWorld and AndroidWorld dynamic benchmark.
3. Beyond main experiments, authors provide sound ablations and failure analyses, clearly attributing gains to new architectural choices.

**Reasons To Reject:**

1. While the paper presents a compelling system design, many of the contributions appear to stem from engineering efforts rather than research. In particular, several design choices seem tailored to the OSWorld benchmark.
2. The proposed Proactive Hierarchical Planning, IMO, involves re-generating the plan after completing each subgoal. However, similar ideas have already appeared in earlier works such as OS-Copilot[1]. Moreover, the multi-step planning capability could also be acquired through training, e.g., OS-Genesis[2], without requiring the added complexity of a multi-agent framework.
3. Regarding experiments: the Structural Grounding Expert updates sheetcells directly, and the Textual Grounding relies on OCR for fine-grained word/line-level localization. This causes the system to circumvent some core challenge of precise pixel-level clicking, which baseline agents like TARS and OpenAI CUA must fight grounding directly. This may introduce bias in the comparison, as Agent S2 is operating with privileged settings (IMO, this discrepancy would likely confer an even greater advantage on benchmarks like Spider2V[3]).

[1] OS-Copilot: Towards Generalist Computer Agents with Self-Improvement

[2] OS-Genesis: Automating GUI Agent Trajectory Construction via Reverse Task Synthesis

[3] Spider2-V: How Far Are Multimodal Agents From Automating Data Science and Engineering Workflows?

---

> ### Author Response · Authors · 2025-06-03
> **Thank you for your review!**
>
> We thank the reviewer for their detailed feedback and for recognizing the Agent S2 architecture, SOTA performance, and soundness of our ablations and analyses. We appreciate the opportunity to address the concerns raised.
>
> **Agent S2 generalizes beyond a particular benchmark.**
>
> We would like to clarify that our contributions are general, and we do not spend engineering efforts tailored specifically to OSWorld for the following reasons:
>
> *Demonstrated Generalization.*
>
> Beyond OSWorld (Ubuntu), Agent S2's architecture demonstrates SOTA performance on WindowsAgentArena (Windows OS) and AndroidWorld (smartphone). This cross-platform success demonstrates that our framework is general and is not overfit or engineered to a particular environment.
>
> *Environment-Agnostic Innovations.*
> - Proactive Hierarchical Planning (PHP) operates in a semantic planning space. All of the Manager’s planning is based on the task description and current state of the desktop, without any human intervention.
> - Mixture of Grounding (MoG) uses experts that are broadly applicable across any operating system. For instance, the Textual Grounding expert employs Optical Character Recognition (OCR), a general technique for extracting text from any screen. Similarly, the Structural Grounding expert provides a programmatic way to interact with tabular data, a common element across various apps and environments. Popular real world computer use tasks that extend beyond OSWorld, such as software engineering and online research, require interaction with textual documents and tabular data, and MoG is consistently applicable in all of these scenarios.
>
> **Regarding Novelty of Proactive Hierarchical Planning.**
>
> We emphasize our final revision will highlight and further compare with OS-Copilot (Wu et al., 2024), OS-Genesis (Sun et al., 2024), AgentStore (Jia et al., 2024), and Spider2-V (Cao et al., 2024) in more detail. However, our Proactive Hierarchical Planning (PHP) offers the following distinct characteristics:
>
> - In Agent S2, the Manager refines its list of subgoals after the termination of each individual subgoal, allowing continuous adaptation at the Manager level. From our understanding of the OS-Copilot (Wu et al., 2024) codebase, their Planner *reactively* replans depending on judgement/repair status. *Proactive* Hierarchical Planning activates the Manager to continually replan, avoiding scenarios where future subtasks become outdated.
> - The hierarchical framework effectively distributes cognitive load across different models. While learning multi-step planning with a single model is promising, as in OS-Genesis (Sun et al., 2024), our ablations demonstrate that no individual model is perfect, and leveraging the strengths of different models for different roles obtains better performance.
>
> **Regarding Fairness of Comparison of MoG.**
>
> Overcoming the bottlenecks of pixel-level grounding is precisely a strength of our framework. It is important to note that the baselines we compare against are also frameworks and not just base models. We show that a monolithic framework like Claude Computer Use that uses the same model for both planning and grounding performs worse than Agent S2. Thus, using separate MoG experts is a deliberate choice to address grounding bottlenecks, while also preserving the reasoning ability of generalists.
>
> MoG is also comparable to previous works such as Omniparser which parses an image to extract interactable elements, Agent-S which uses OCR to augment its accessibility tree input, as well as standard Set-of-Mark approaches. We emphasize that Agent S2 does not use any privileged information from the environment. In fact, while a range of previous works use additional information like Accessibility Trees and HTML content, our approach relies purely on screenshot input.
>
> **Response for Questions to Authors**
>
> MoG Routing Strategy: The selection of the grounding expert is dynamically decided by the Worker module at each execution step. This is not enforced through rules. The key difference in AgentStore (Jia et al., 2024) is that their MetaAgent preselects application-specific agent(s) to perform a full task/subtask based on special tokens. In contrast, Agent S2 proposes a small set of very general classes of grounding experts: Visual, Textual, and Structural, that are always available and dynamically selected at the step level.
>
> We reference Figure 4 as an example. For the subgoal: “select last paragraph”, the Worker initially tries the visual grounding expert by calling the agent.click() action. Immediately after, it decides to use the textual grounding expert by calling the agent.select_span() action after observing that the initial click did not highlight the paragraph properly. We do not enforce any heuristics or restrictions, such as assigning an expert per subtask. Instead, the ability of Agent S2 to reason and reflect on the desktop state is fully responsible for which grounding expert is used at each step.

---

> > ### Author Response · Authors · 2025-06-03
> > **Citations**
> >
> > 1. Cao, R., Lei, F., Wu, H., Chen, J., Fu, Y., Gao, H., Xiong, X., Zhang, H., Mao, Y., Hu, W., Xie, T., Xu, H., Zhang, D., Wang, S., Sun, R., Yin, P., Xiong, C., Ni, A., Liu, Q., … Yu, T. (2024). Spider2-V: How far are multimodal agents from automating data science and engineering workflows? [Preprint]. arXiv. https://arxiv.org/abs/2407.10956
> >
> > 2. Jia, C., Luo, M., Dang, Z., Sun, Q., Xu, F., Hu, J., Xie, T., & Wu, Z. (2024). AgentStore: Scalable integration of heterogeneous agents as specialized generalist computer assistant [Preprint]. arXiv. https://arxiv.org/abs/2410.18603
> >
> > 3. Sun, Q., Cheng, K., Ding, Z., Jin, C., Wang, Y., Xu, F., Wu, Z., Jia, C., Chen, L., Liu, Z., Kao, B., Li, G., He, J., Qiao, Y., & Wu, Z. (2024). OS-Genesis: Automating GUI agent trajectory construction via reverse task synthesis [Preprint]. arXiv. 4. https://arxiv.org/abs/2412.19723
> >
> > 4. Wu, Z., Han, C., Ding, Z., Weng, Z., Liu, Z., Yao, S., Yu, T., & Kong, L. (2024). OS-Copilot: Towards generalist computer agents with self-improvement [Preprint]. arXiv. https://arxiv.org/abs/2402.07456

---

> > ### Comment · Reviewer_Ygqb · 2025-06-07
> >
> > Thanks for your response, I think my concerns are partially addressed, and revise my rating 6 -> 7.

---

> > > ### Author Response · Authors · 2025-06-08
> > >
> > > Thank you for your feedback and score update! Please let us know if you have any more questions.

---

### Official Review · Reviewer_Cqe8 · 2025-05-12

**Rating:** 8
**Confidence:** 4
**Ethics Flag:** 1

**Summary:**

This paper presents Agent S2, a modular framework for GUI agents that integrates generalist and specialist models to improve performance on complex GUI tasks. The framework introduces two key components: a Mixture of Grounding strategy that delegates GUI localization to specialized experts (visual, textual, and structural), and Proactive Hierarchical Planning, which enables dynamic replanning based on evolving observations. Agent S2 achieves good results on multiple real-world benchmarks including OSWorld, WindowsAgentArena, and AndroidWorld.

**Questions To Authors:**

1. How will you handle the deployment or substitution of specialist modules (e.g., grounding experts) in environments where those specific tools may not be available?

2. Additionally, the Agent S2 framework involves a fairly complex architecture with multiple coordinated modules. How do you envision this system being deployed or maintained in real-world settings, particularly with respect to integration complexity, latency, and robustness across components?

**Reasons To Accept:**

- The paper proposes a new GUI agent, Agent S2, that integrates generalist planning with specialist grounding components. This includes a Mixture of Grounding (MoG) system and Proactive Hierarchical Planning, addressing bottlenecks in GUI agents.

- Agent S2 achieves state-of-the-art results across multiple real-world benchmarks—OSWorld, WindowsAgentArena, and AndroidWorld—outperforming baselines such as Claude CUA, UI-TARS, and GPT-4o-based systems.

- The paper includes detailed ablation studies and failure case breakdowns, giving transparency into the contribution of each component.

**Reasons To Reject:**

- I do not think this paper have any good reason to be rejected, altough I think the only part can be improve is the system’s multiple moving parts: manager, worker, three grounding experts, and multiple LLMs—add complexity. This may make deployment or real-time interaction more challenging without significant engineering effort.

---

> ### Author Response · Authors · 2025-06-03
> **Thank you for your review!**
>
> We sincerely thank the reviewer for their positive assessment, strong support for our paper, and insightful questions regarding the practical deployment of Agent S2. We are encouraged by the reviewer's recognition of our state-of-the-art results and the transparency of our analyses.
>
> We will address the questions concerning the deployment and modularity of Agent S2:
>
> **Regarding the handling of specialist modules in diverse environments and their potential unavailability.**
>
> The Worker module dynamically selects the most appropriate grounding expert based on its reasoning. This implies that if a particular type of interaction is not available or the agent encounters a failure, the Worker will naturally fall back to alternative actions. This dynamic selection allows flexible extension and replacement of various experts without breaking the system.
>
> Additionally, the grounding experts used in Agent S2 are broadly generalizable and applicable across various different applications and operating systems.
>
> - The Visual Grounding expert relies on screenshot input and language descriptions to locate elements, a general approach applicable to any GUI. This can be implemented with smaller models (UI-TARS-7B, UGround-V1-7B), which are cheaper to deploy.
> - The Textual Grounding expert, which utilizes Optical Character Recognition (OCR), can identify and locate text on any screen, irrespective of the underlying OS or application. Various open source softwares and libraries like pytesseract, paddle-ocr, as well as services from providers like Google, Amazon, Mistral etc. can be used for the textual grounding expert.
> - The Structural Grounding expert can work with any sheet-based application. While we use unotools to interface with Libreoffice calc, other sheet–based applications provide alternatives like Openpyxl for Excel, Google Apps Script for Google Sheets, AppleScript for Numbers etc.
>
> **Regarding the deployment and maintenance of Agent S2 in real-world settings, particularly integration, robustness and latency.**
>
> - The modular design of separating responsibilities across the Manager, Worker, and distinct Grounding experts, enhances the system's architectural robustness. This means replacing or ablating individual modules can be done without system failures.
> - The modular design also helps reduce work for future development, as users can easily plug-and-play newer models, allowing the agent to evolve with minimal changes to the framework and codebase.
> - Furthermore, both generalist planning models (e.g., Claude, GPT-4o) and specialist grounding models (e.g., UI-TARS) are currently supported via serverless APIs, which alleviates the difficulties of local or user-managed deployments.
> - We acknowledge that latency is a critical consideration for real-time interaction in real-world deployments. The current work has focused on establishing the efficacy of the Agent S2 framework and achieving state-of-the-art performance on complex, long-horizon tasks. While our work does not focus on latency, users of Agent S2 can easily swap in smaller proprietary or open-source models to decrease latency.

---

> > ### Comment · Reviewer_Cqe8 · 2025-06-05
> >
> > Thanks for your reply, I will maintain my positive score:)

---

> > > ### Author Response · Authors · 2025-06-08
> > >
> > > Thank you for your feedback and your support!

---

### Official Review · Reviewer_u6Sn · 2025-05-17

**Rating:** 6
**Confidence:** 3
**Ethics Flag:** 1

**Summary:**

This paper presents Agent S2, a compositional generalist-specialist framework for computer use agents. The framework is illustrated in Figure 2, which comprises generalist planning modules, a manager, workers, and specialist grounding experts to handle complex computer use tasks. The authors conducted experiments on the OSWorld benchmark, consisting of 369 computer use tasks across multiple categories. The results show Agent S2 with Claude-3.7-Sonnet performs best compared to other baselines.

**Reasons To Accept:**

- The proposed framework with a combination of multiple models for visual, textual, and structure grounding experts together with a strong LLM, shows improvement on the computer use tasks
- Good experiments with ablation studies and error analysis
- Potential to generalize to smartphone use

**Reasons To Reject:**

The proposed framework is a combination of various models with a highly abstract description. It is not clear what the key innovation of the proposed work is.
- Lack of detailed ablation studies to show the key contribution of each model in the framework, especially if we replace Claude with an open-source LLM.

---

> ### Author Response · Authors · 2025-06-03
> **Thank you for your review!**
>
> Thank you for your insightful review and for recognizing the potential of Agent S2. We appreciate your positive comments on the framework's demonstrated improvements, the quality of our experiments and ablation studies, and its generalization potential. We welcome this opportunity to clarify the key innovations of our work and provide further details on our ablation studies. As per the reviewer’s request we also report additional experimental performance with an open source planner model.
>
> **Regarding key innovation.**
>
> - Agent S2's primary innovation is the empirical demonstration that a compositional framework that combines Generalist models for planning/reasoning with Specialist models for grounding can outperform stronger monolithic models.
> - We present a concrete description of the hierarchical architecture used in Agent S2 (in Section 3), consisting of a Manager module ($M$), Worker module ($W$), and Grounding experts ($G_i$).
> - We also introduce two key strategies, the Proactive Hierarchical Planning (Section 3.2) and Mixture of Grounding (Section 3.1) and quantify the benefits of these strategies through ablation studies in Figure 5.
> - Our results show that Agent S2 achieves State-of-the-art results across two computer use benchmarks and 1 smart phone use benchmark
> - We will open-source the code of our framework after the revision for complete reproducibility..
>
> **Regarding ablation on base models.**
>
> We would like to clarify that our paper ablates on different foundation models as the common backbone for Manager and Worker. The results in Table 2 show the categorized performance of using different generalist models as the Manager and Worker, demonstrating that Agent S2 amplifies the performance of all underlying models.
>
> | Planner and Worker Model       | 15-Step Success Rate on OSWorld (%) |
> |--------------------------------|-------------------------------------|
> | Qwen2.5-VL-72B-Instruct         | 18.29                              |
> | GPT-4o                          | 21.12                              |
> | Claude-3.5-Sonnet (new)        | 24.50                              |
> | Claude-3.7-Sonnet              | **27.04**                              |
>
> Furthermore, as per your suggestion, we run a full 15 step evaluation on OSWorld using the open-source Qwen2.5-VL-72B-Instruct as the backbone planning model. Agent S2 with Qwen2.5-VL-72B-Instruct obtained a success rate of 18.29%, more than doubling the performance over the reported baseline Qwen2.5 model score of 8.83%. Our framework still demonstrates large improvements, even when plugging in open-source VLMs.
>
> We would also like to point out that our paper already ablates on different open-source models as the visual grounding expert. Our ablation study on the Visual Grounding expert (Figure 6) illustrates how smaller, open-source specialist models like UI-TARS-7B-DPO (29.23% success rate) and UGround-V1-7B (24.61% success rate) can match and even outperform a large generalist model like Claude-3.7-Sonnet (24.61% success rate) when employed as a dedicated visual grounding expert within our modular Agent S2 framework.
> | Expert Model (Claude-3.7-Sonnet as Manager and Worker) | Success Rate on OSWorld Validation Set (n=65) (%) |
> |--------------------------------------------------------|---------------------------------------------------|
> | UI-TARS-72B-DPO                                        | **30.77**                                            |
> | UI-TARS-7B-DPO                                         | 29.23                                            |
> | UGround-V1-7B                                          | 24.61                                            |
> | Claude-3.7-Sonnet                                      | 24.61
>
> Our focus in this study was discovering the framework design methodology that works best, and comparing against the best available baselines to demonstrate that well motivated, open-source frameworks can outperform closed models and frameworks. This is why we chose to use strong generalist models in our study. Exploring the performance trade-offs with a wider range of open-source models represents an important avenue for future work and our open-source framework allows future researchers to easily compare and conduct these experiments.

---

> > ### Author Response · Authors · 2025-06-08
> >
> > Thank you for your feedback and review of our work. We have posted a reply that clarifies our innovations and shows the open-source model results you suggested. We would be glad to hear your thoughts and whether we have fully addressed all your questions and concerns!

---

### Official Review · Reviewer_YT8u · 2025-05-23

**Rating:** 6
**Confidence:** 4
**Ethics Flag:** 1

**Summary:**

This paper proposes Agent S2, a GUI agent framework with two main components: Proactive Hierarchical Planning, which replans at every step, and a Mixture-of-Grounding Experts to achieve more precise GUI grounding. The method is tested in three GUI environments: OSWorld, WindowsAgentArena, and AndroidWorld, and achieves state-of-the-art performance.

**Questions To Authors:**

None.

**Reasons To Accept:**

- The paper is overall well written, accompanied by clean and intuitive figures.
- The proposed “Proactive Hierarchical Planning,” which replans at every step, is sound, as it is often impossible to generate a perfect plan at the beginning in a partially observable environment.

**Reasons To Reject:**

The major weakness of this paper is the significant lack of details and experiments, elaborated below:
- **Unclear whether the improvement comes from a better framework design or simply from using a better base LLM**: Among the baselines in Table 1, only Claude Computer Use (CCU) shares the same base LLMs (Claude-3.5-Sonnet (new) and Claude-3.7-Sonnet) as the proposed method, while most other baselines use GPT-4o, which performs worse than Claude-3.5-Sonnet (new) and Claude-3.7-Sonnet as shown in Table 2. Instead of a single “Method” column, I would suggest the authors split this into separate “Framework” and “Model” columns for a more rigorous comparison.
- **Unclear whether the improvement comes from a better framework design or from a more tailored action space**: Table 8 shows that the authors added several actions to make it easier for the agent to interact with the environment. Are the current baselines using the same action space? How does this tailored action space affect agent performance in OSWorld, WindowsAgentArena, and AndroidWorld?
- **Lack of fine-grained baseline comparison**: While I appreciate the detailed breakdown of success rates in Table 2, it provides little insight into how Agent S2 performs compared to baselines at a finer level. I would suggest the authors compare the categorized success rates with those of the baselines as well.
- **Insufficient baselines for some environments**: There are currently only 2-3 baselines for WindowsAgentArena and AndroidWorld. I would suggest the authors include more baselines for a more comprehensive comparison.
- **Unclear cost implications**: Agent S2 replans at every step, which raises concerns about additional cost. How expensive is Agent S2 compared to the baselines (e.g., in terms of the number of LLM calls, number of input/output tokens, and time to finish a task)?
- **Limited novelty**:
  - The Proactive Hierarchical Planning component is essentially just replanning at every step given new observations.
  - The “Mixture-of-Grounding-Experts” approach is equivalent to the tool-using paradigm, where each expert functions as a tool. In this paper, three tools are considered: a VLM for visual grounding, `tesseract` for OCR, and `unotools` to interact with OpenOffice.org/LibreOffice. The `unotools` is tailored to the OSWorld environment, which may explain the strong performance on office-related tasks (Table 2 and L271).
- Other minor concerns:
  - Exact prompts and model versions (GPT-4o and Claude models) should be included for reproducibility.
  - Details on how the error analysis was conducted should be provided.

In summary, I am concerned that the lack of analysis and experimentation makes it difficult to identify the true source of accuracy gains, which may arise from trivial factors.

---

> ### Author Response · Authors · 2025-06-03
> **Thank you for your review!**
>
> Thank you for your detailed review of our work, which has prompted us to clarify several aspects of our paper and highlight the contributions more effectively. We will address the concerns below:
>
> **Source of improvement between framework and base models**
>
> We wish to address the misunderstanding about stronger base models, and clarify that the evidence strongly points to the Agent S2 framework improving the task success performance. Below we present a table showing the success rates of various frameworks using screenshot input, grouped by the underlying base model on OSWorld.
>
> | Model                   | Framework               | 15-Step (%) | 50-Step (%) |
> |------------------------|-------------------------|---------------------|---------------------|
> | GPT-4o                 | Aria-UI                 | 15.2                |                   |
> | GPT-4o                 | Aguvis-72B              | 17.0                |                    |
> | GPT-4o                 | Agent S2                | **21.1**                | **26.62**               |
> | Claude-3.5-Sonnet (new)| Claude Computer Use | 14.9                | 22.0                |
> | Claude-3.5-Sonnet (new)| Agent S2                | **24.5**                | **33.7**                |
> | Claude-3.7-Sonnet      | Claude Computer Use | 15.5                | 26.0                |
> | Claude-3.7-Sonnet      | Agent S2                | **27.0**                | **34.5**                |
>
> - The table shows that Agent S2, with the same base model outperforms the previous SOTA frameworks.
> - Agent S2 w. Claude-3.5-sonnet (new) outperforms Claude Computer Use framework with the better Claude-3.7-sonnet. This further validates that Agent S2 improves performance by amplifying the abilities of the base model.
>
> We used Table 1 to compare with SOTA baselines, irrespective of input modalities and special fine-tuning. However, we will include the above table in our revision to directly compare frameworks grouped by the underlying base model.
>
> **Generality of action space and framework contributions**
>
> - We point out that action spaces are an integral part of a framework. All baselines used in this work have their own action space. The purpose of the framework and its action space is to elicit the abilities of the base model for the provided task.
> - We also clarify that our action space is not tailored specifically for the OSWorld benchmark. Our action space generalizes to various applications (Web navigation, desktop software, text editors, etc.) and operating systems (Ubuntu, Windows, and MacOS).
> - The actions set_cell_values and highlight_text_span, are interfaces to our MoG method. Their functionality is enabled by the grounding experts.
> - The same action space is used in OSWorld and WindowsAgentArena underscoring the general effectiveness of our framework and action space. For AndroidWorld, we adapt actions like swipe, back, home, and long_press which also exist in the baselines that we compared against.
>
> **Fine-grained baseline comparisons**
>
> Amongst the state-of-the-art baselines we compare with, only Aria-UI and Agent S report their categorized success rate, specifically on the 15-step benchmark. We will include this table in the revised version.
> | Input Modality               | Method                          | OS     | Daily  | Office | Professional | Workflow | Overall |
> |-----------------------------|----------------------------------|--------|--------|--------|--------------|----------|---------|
> | Screenshot Only             | Aria-UI (GPT-4o)                 | 25.00  | 25.71  | 9.58   | 20.41        | 8.55     | 15.15   |
> | Screenshot + Atree | Agent S (GPT-4o)             | 45.84  | 27.06  | 13.00  | 36.73        | 10.53    | 20.58   |
> | Screenshot Only             | Agent S2 (GPT-4o)                | 37.50  | 25.57  | 11.96  | 48.98        | **10.89**    | 21.12   |
> | Screenshot Only             | Agent S2 (Claude-3.5-Sonnet (new))| **45.84** | **35.82**  | 15.52  | 57.14        | 5.00     | 24.50   |
> | Screenshot Only             | Agent S2 (Claude-3.7-Sonnet)     | 41.67  | 35.62  | **20.51**  | **65.31**        | 5.94     | **27.04**   |
>
> **Baseline comparisons in other benchmarks**
>
> For WindowsAgentArena (WAA) and AndroidWorld (AWD), we already report comparisons with the previous SOTA frameworks. The cited baselines on WAA are the only reported scores at the time of the submission, to the best of our knowledge. For AWD, we already picked the three top-performing baselines for comparison, but we will include following additions in our revision.
>
> | Framework       | SR (%) |
> |----------------|------------------|
> | Agent S2       | **54.3**             |
> | UI-TARS        | 46.6             |
> | Aria-UI        | 44.8             |
> | UGround        | 44.0             |
> | Ponder & Press | 34.5             |
> | AndroidWorld   | 30.6             |
> | InfiGUIAgent   | 9.0              |

---

> > ### Author Response · Authors · 2025-06-03
> >
> > **Cost implications**
> >
> > We wish to clarify the misunderstanding about Agent S2’s replanning process. Agent S2 does not replan (generate a new subgoal list) after each step but rather after the termination of each subgoal. We report the cost statistics of Agent S2 in the below table and compare it with Agent S. We use the same GPT-4o model used by Agent S, for a fair comparison. To the best of our knowledge, the other baselines do not report cost statistics.
> >
> >
> > | Agent  (15-steps)        | Avg LLM Calls/Task | Avg Time/Task (s) | Avg Input Tokens/Step | Avg Output Tokens/Step |
> > |--------------------|--------------------|----------------------|------------------------|-------------------------|
> > | Agent S* | 26.74              | 494.78               | 18393.30               | 216.98                  |
> > | Agent S2 | 27.98              | 568.38               | 8037.13                | 267.94                  |
> >
> > Even though Agent S2 replans proactively, while Agent S replans reactively, we see that the increase in average Manager+Worker calls per task is minimal. We attribute this to Agent S2’s effective task completion and higher success (even though there are more Manager LM calls, it iteratively generates clearer and more concise plans, which lead to better subtask completion and less Worker LM calls). Furthermore, the amount of input tokens per step in Agent S2 is actually less than half as much as in Agent S due to both the screenshot and accessibility tree being used as the input for Agent S.
> >
> > *\*Calculations for Agent S based on their publicly reported trajectories which include token and time logs.*
> >
> > **Novelty of framework**
> >
> > - We reiterate our paper's central contribution: *An empirical demonstration that a modular framework combining generalist and specialist models can surpass stronger, single monolithic model frameworks.* This is evidenced by Agent S2 with Claude-3.5-Sonnet outperforming the monolithic Claude Computer Use with the more advanced Claude-3.7-Sonnet. Our ablation studies (Figure 5) further show that this modular approach scales more effectively with increased time steps and computation.
> > - We'd like to clarify that *Agent S2 does not replan at every step*, but rather operates using a hierarchical structure. A comprehensive plan (subgoal list) is updated only after each subgoal is completed, not after every step. At the lower, step-by-step level, the worker module reasons and acts on a single subgoal for multiple steps until termination. This hierarchical approach offers two benefits: it keeps the agent focused on the broader task by completing subgoals sequentially, and it allows the agent to adapt to new observations by updating the subgoals after each one is finished.
> > - *The Mixture of Grounding (MoG) is an abstraction compatible with the classic tool-use paradigm; however, the experts are not tools themselves*. Our framework's action space (tools) can function independently of the grounding experts. We demonstrate both this independence and the effectiveness of MoG in the ablation study in Figure 5. MoG represents an interface adaptable to any action space that enables the Worker to generate all actions and element descriptions purely in a semantic space, relieving it of the cognitive load of grounding.
> >
> > **Unotools**
> >
> > We use unotools as the backend for the Structural Grounding expert to interact with LibreOffice. While unotools reflects a specific software choice, the concept of the Structural Grounding expert, which involves interacting with applications via semantic descriptions of updates to structured data ({"cell": "value"} mappings), is general and applicable to any spreadsheet software or structured UI element across all operating systems. Other sheet applications provide alternatives like Openpyxl for Excel, Google Apps Script for Sheets, AppleScript for Numbers etc. Similarly, our Textual grounding expert can be generally applicable to any operating system, application, or website and can be implemented using various available OCR softwares and tools.
> >
> > **Prompts and Models**
> >
> > We will open-source our code and all prompts for reproducibility and access. However, we appreciate the review’s suggestion and will incorporate the prompts and model versions in our appendix in the revision. The versions of models used are: gpt-4o-2024-08-06, claude-3-5-sonnet-20241022, and claude-3-7-sonnet-20250219.
> >
> > **Error analysis**
> >
> > We conduct error analysis on a subset of 65 samples from OSWorld through qualitative evaluation. For each example, we report the theme of the first major cause of failure. The descriptions of the themes can be found in Section 4.4. We appreciate the suggestion of incorporating additional details of the error analysis and will add it in our revision.

---

> ### Comment · Reviewer_YT8u · 2025-06-07
> **Follow-up Discussion**
>
> I thank the authors for their efforts in addressing my concerns. I would like to follow up on a few remaining points below:
>
> **Regarding “Generality of Action Space and Framework Contributions”**
>
> I kindly disagree. Unless the manipulation of the action space is a central contribution of the paper, it should remain consistent across baselines, or at the very least, unaltered from the target environment, to ensure fair comparisons. Otherwise, it becomes too easy for the agent designer to introduce new actions that simplify the environment, potentially provide an unfair advantage and trivially improve performance. I strongly encourage the authors to include a more detailed ablation study on the added actions to clarify their impact.
>
> **Regarding “Baseline Comparisons in Other Benchmarks”**
>
> Thank you for the additional details. However, as previously noted, comparing agent frameworks without fully disclosing the underlying LLMs is strongly discouraged. For example, it is unsurprising that Agent S2 paired with Claude-3.7-Sonnet outperforms another framework using a weaker LLM such as LLaMA-2-7B. Without controlling for backbone differences, such comparisons are difficult to interpret.
>
> **Regarding “Novelty”**
>
> I remain unconvinced by the claimed novelty of the proposed framework. The "Proactive Hierarchical Planning" mechanism appears to be standard replanning upon receiving new observations, while the "Mixture of Grounding" seems to be a rebranding of the existing tool-use paradigm.

---

> > ### Author Response · Authors · 2025-06-08
> > **Follow-up Response**
> >
> > **Re: Action Space**
> >
> > We believe there is a misunderstanding regarding the action space. All computer use benchmarks we report require executing Python code to interact with the underlying UI environments—a protocol consistently followed by both prior work and our own.
> >
> > The action space of computer use agents is a language-based abstract layer between LLM agents and direct UI interaction, which is an integral part of agentic frameworks to effectively elicit LLM’s agentic capabilities. Evolving how to better elicit LLM’s agentic capabilities itself is scientifically valuable, as all reasoning occurs through free-form language generation, and the language-based actions are the primary means by which LLM agents interact with their environment.
> >
> > Insisting that this high level action space remain identical across all baselines contradicts established practice in agentic AI research and could hinder scientific progress, as prior language-based action spaces are often suboptimal to elicit LLM’s agentic capabilities. The strong, SOTA baselines we compare against have their own language-based action spaces better suited to their frameworks, for instance:
> >
> > - **Agent S**: Utilizes an ID-based action space specifically optimized for its dual-modality input (screenshots and accessibility trees).
> > - **Claude Computer Use**: Introduces custom action spaces incorporating advanced command-line interactions through Bash tools and text-editing capabilities for file operations.
> > - **UI-TARS**: was trained on its own action space that includes a call_user function for data annotation during training and for infeasible task prediction during inference.
> >
> > Furthermore, we demonstrated the general applicability of our framework with a universal action space across multiple operating systems such as Ubuntu, Windows, and Android.
> >
> > **Re: Baseline Comparison**
> >
> > There appears to be a significant misunderstanding. We have explicitly mentioned the underlying LLMs and provided the appropriate citations for all comparisons in our original paper (Table 1). In our previous response, we also included a structured table grouping results by base model, demonstrating that Agent S2 achieves the best results within each group using the same LLM backbone, such as GPT-4o, Claude-3.5, and Claude-3.7. Notably, Agent S2 with an inferior LLM (Claude-3.5 or GPT-4o) even outperforms Claude Computer Use (CCU) paired with a stronger LLM (Claude-3.7).
> >
> > To reiterate our experimental setup and key observations:
> >
> > - Agent S2 outperforms state-of-the-art (SOTA) baselines using the same underlying models. This is demonstrated in Table 1 of our paper, where the backbone LLM for each method is clearly specified. These results were further clarified in our rebuttal with an additional structured comparison.
> > - We analyze the impact of different base models. Table 2 systematically examines how Agent S2 performs using a variety of LLMs, showing how model capability affects outcomes across task categories.
> > - We validate generalization to other benchmarks. For WindowsAgentArena and AndroidWorld, we compare Agent S2 to the publicly reported SOTA at the time of submission. These experiments were designed to assess generalizability. The effects of model differences had already been rigorously controlled and analyzed in the first two studies.
> >
> > We have not reported any comparisons against baselines using LLaMA-2-7B. If this was inferred, it likely stems from a misunderstanding. We take great care to ensure that all reported comparisons are either model-controlled or transparently documented when based on external reports.
> >
> > **Re: Novelty**
> >
> > We respectfully highlight that perceptions of "novelty" are inherently subjective and may occasionally reflect some personal biases rather than an objective assessment of contributions.
> >
> > Our key scientific contribution is the robust empirical demonstration that a compositional Generalist-Specialist framework, exemplified by Agent S2, significantly outperforms monolithic frameworks, even those leveraging stronger underlying models (e.g., Agent S2 with weaker models outperforms Claude computer use with stronger Claude-3.7-sonnet).
> >
> > Based on this framework, we made two further architectural contributions. First, Mixture of Grounding (MoG) empowers the generalist model to reason in semantic terms. By offloading the specialized task of generating precise, coordinate-based grounding to expert components, MoG distinctly enhances the overall efficiency and effectiveness of the framework. Second, Proactive Hierarchical Planning (PHP) enables the agent to strategically replan at the subgoal level, maintaining task coherence while dynamically adapting to new observations at different levels.
> >
> > Ultimately, our philosophy prioritizes simplicity and effectiveness: if a method works well and achieves state-of-the-art results across multiple benchmarks and models, simplicity itself is valuable.

---

> > > ### Comment · Reviewer_YT8u · 2025-06-09
> > > **Follow-up Discussion**
> > >
> > > **Regarding “Generality of Action Space and Framework Contributions”**
> > >
> > > Thank you for the detailed clarification. To avoid any further miscommunication, I will try to summarize my concern more clearly:
> > >
> > > 1. I understand that OSWorld and WindowsAgentArena support arbitrary code execution as actions, meaning the action space can theoretically include any valid program. That said, both environments still define finite, default action sets. For OSWorld, this is the computer_13 set (as listed in Table 8 of the paper), and for WindowsAgentArena, it corresponds to the methods in their Computer class (Table 6). In contrast to the authors’ previous response, AndroidWorld does not support arbitrary code execution, but instead provides a fixed set of predefined actions (see Sections 3.2 and C.2 in the AndroidWorld paper).
> > > 2. According to the paper (Line 484), Agent S2 introduced additional actions *“to make it easier for the agent to interact with the environment.”* From what I understand, `highlight_text_span` and `set_cell_values` are among the newly added actions, while the rest are for compatibility with the different environments.
> > > 3. My core question remains: **How does the inclusion of these additional actions impact Agent S2’s performance?**
> > >
> > > One way to address this would be through an ablation study on the added actions, which I suggested earlier. As of now, this point still appears to be unaddressed.
> > >
> > > To be clear, I do not object to frameworks having their own action spaces per se, but when new actions are introduced that may simplify the task, I believe it is important to isolate their contribution through ablations, especially if the action design is not a central methodological focus.
> > >
> > > **Regarding “Baseline Comparisons in Other Benchmarks”**
> > >
> > > Thank you for the response. However, my earlier comment referred specifically to the table shared in the previous response (reattached below), where the underlying base LLMs are not disclosed:
> > >
> > > | Framework      | SR (%) |
> > > | -------------- | ------ |
> > > | Agent S2       | 54.3   |
> > > | UI-TARS        | 46.6   |
> > > | Aria-UI        | 44.8   |
> > > | UGround        | 44.0   |
> > > | Ponder & Press | 34.5   |
> > > | AndroidWorld   | 30.6   |
> > > | InfiGUIAgent   | 9.0    |
> > >
> > > As a general guideline, it would be helpful if each framework listed in the table also includes the base LLM used. This helps the reader better understand whether performance differences may stem from the underlying model or the framework itself.
> > >
> > > **Rating Update**
> > >
> > > With the additional experiments showing that (1) Agent S2 performs competitively even with the GPT-4o backbone (outperforming CCU), and (2) other concerns, including fine-grained comparisons, more baselines for WindowsAgentArena and AndroidWorld, and cost breakdowns, have been addressed, I am raising my score from **3 to 4**.
> > >
> > > I would be happy to consider a further increase in score once an ablation study on the added actions is included.

---

> > > > ### Author Response · Authors · 2025-06-09
> > > >
> > > > **Regarding the grounding expert interface actions.**
> > > >
> > > > We appreciate your clarification. First, we would like to reiterate that those two actions are the agent’s interfaces for the Structural and Textual Grounding Experts. They generalize across multiple domains and applications. We would like to refer to Sec. 4.3 in the paper for the ablation of Mixture of Experts by removing all three experts from Agent S2 which also ablates the two actions.
> > > >
> > > > To provide the exact ablation as requested, we reran Agent S2 w/ Claude-3.7-sonnet on the same subset of OSWorld that we used to conduct our previous ablations on the Visual Grounding models, without the two actions (15 steps): the full Agent S2’s score is 30.77%, and removing those two actions results in a score of 29.23%.
> > > > Meanwhile, the experimental results we mentioned in the paper in Section 4.3 on line 289 provide a more comprehensive and fine-grained subtask-level inspection of ablating the two experts:
> > > >
> > > > > To further study the benefit of individual experts, we extract a subset of OSWorld examples where the agent routes actions to the Textual or Structural Grounding expert. We then re-evaluate those examples without the corresponding expert to measure their impact on successful subtask completion. When removing the textual grounding expert, the subtask success rate drops from 70.6% to 65.2%, and when removing the structural grounding expert, the subtask success rate drops from 73.7% to 69.4%.
> > > >
> > > > **Regarding the table.**
> > > >
> > > > Regarding the table included in our rebuttal, we aimed to reassure you that additional baselines will be incorporated in the revised version of the paper as requested. The LLM backbone-related ablations were explicitly detailed in other sections addressing your queries about the LLM backbones.
> > > >
> > > >  We would also like to reiterate that the experiments on WindowsAgentArena and AndroidWorld are conducted for the purpose of demonstrating generalization to other benchmarks (the ablations are done on OSWorld). In Table 4 of the paper, we have clearly demonstrated the underlying base models for baselines. In Sec. 4.1 Implementation Details in the paper, we also stated that we use Claude-3.7-sonnet for the best results on those benchmarks.
> > > > We appreciate your feedback and will explicitly clarify and include detailed information about underlying LLMs for all results in the revision to further enhance transparency and clarity.

---

> > > > > ### Comment · Reviewer_YT8u · 2025-06-10
> > > > > **Follow-up Discussion**
> > > > >
> > > > > Thank you for the clarification. Based on the new ablation results, it appears that the added actions do not significantly inflate performance, which addresses my remaining concern. I appreciate the authors’ continued effort in providing detailed follow-ups throughout the discussion. I will raise my rating from 4 to 6.

---

> > > > > > ### Author Response · Authors · 2025-06-10
> > > > > >
> > > > > > Thank you for your thoughtful review, the score update, and your engagement in our detailed follow-up discussions!

---

### Decision · Program_Chairs · 2025-07-08

**Decision:**

Accept

**Comment:**

This paper presents a modular approach that decomposes cognitive tasks across generalist planning modules and specialist grounding experts to improve performance on complex GUI tasks. Reviewers recognized the novelty of the Mixture-of-Grounding strategy and Proactive Hierarchical Planning, and noted the strong empirical results demonstrating state-of-the-art performance across multiple benchmarks. The framework’s ability to generalize across different operating systems and applications was also highlighted positively. Concerns were raised about the clarity regarding the source of improvements and the impact of a tailored action space that may simplify tasks relative to baselines. Some reviewers questioned the novelty of the planning approach, noting similarities to prior work, and noted the complexity and engineering effort involved in the multi-component system. Authors clarified that improvements hold when controlling for base models, providing ablation studies showing the contribution of individual components and added actions, and emphasizing the generality of the action space and grounding experts. Reviewers appreciated these clarifications, with some raising their scores accordingly. Overall, the paper was viewed as a strong contribution with sound methodology and significant empirical validation, with some reservations about novelty and complexity. I agree with reviewer consensus towards acceptance given the demonstrated advances and thorough author responses.